# Manipulation of sterol homeostasis for the production of 24-epi-ergosterol in industrial yeast

Yiqi Jiang [1], Zhijiao Sun[1], Kexin Lu[1], Zeyu Wu[1], Hailong Xue[1], Li Zhu[1], Guosi Li[1], Yun Feng[1], Mianbin Wu[1], Jianping Lin [1] ✉, Jiazhang Lian [1,2,3] ✉ & Lirong Yang [1,2]

Brassinolide (BL) is the most biologically active compound among natural brassinosteroids. However, the agricultural applications are limited by the extremely low natural abundance and the scarcity of synthetic precursors. Here, we employ synthetic biology to construct a yeast cell factory for scalable production of 24-epi-ergosterol, an un-natural sterol, proposed as a precursor for BL semi-synthesis. First, we construct an artificial pathway by introducing a $\Delta^{24(28)}$ sterol reductase from plants (DWF1), followed by enzyme directed evolution, to enable *de novo* biosynthesis of 24-epi-ergosterol in yeast. Subsequently, we manipulate the sterol homeostasis (overexpression of *ARE2*, *YEH1*, and *YEH2* with intact *ARE1*), maintaining a balance between sterol acylation and sterol ester hydrolysis, for the production of 24-epi-ergosterol, whose titer reaches to $2.76 \, g \, L^{-1}$ using fed-batch fermentation. The sterol homeostasis engineering strategy can be applicable for bulk production of other economically important phytosterols.

Brassinosteroids (BRs), now recognized as the sixth class of plant hormones[1], involved in the regulation of many vital plant physiological activities, such as cell elongation, cell division, seed germination, immunity, and reproduction[2]. Brassinolide (BL), representing the most biologically active compound among natural brassinosteroids, has great potential for agricultural and industrial applications[3]. However, the natural abundance of this growth-promoting phytohormone is extremely low. For example, Grove et al. only isolated 4 mg BL from 40 kg of bee-collected rape pollen[4]. Thus, scalable production strategies other than plant extraction should be established for practical applications.

Considering the structural complexity of BRs, semi-synthesis based on naturally available sterols has been proposed and attempted[5]. Currently, only two relatively low active BRs (Supplementary Table 1), 24-epi-brassinolide (EBL) and 28-homo-brassinolide (HBL), which can be synthesized from ergosterol[6] and stigmasterol[7], have been commercially produced. As for the production of BL,

crinosterol, isolated from a scarce invertebrate inhabiting the ocean (crinoidea), has been proposed as the synthetic precursor (Fig. 1a). Unfortunately, the availability of crinosterol is no better than BL[8,9]. In recent years, the development of synthetic biology opens up the possibility for bulk production of complex natural products (e.g., BL or synthetic precursors) using microbial cell factories. Without the elucidation of BL or crinosterol biosynthetic pathways (Supplementary Fig. 1), fermentative production of sterol molecules with similar structures as crinosterol becomes a promising alternative for the semi-synthesis of BL. By comparing the side chains of ergosterol, stigmasterol, and crinosterol, we propose 24-epi-ergosterol as an alternative precursor for BL synthesis (Fig. 1a). Although the biosynthesis of 24-epi-ergosterol has not been found in nature, the expansion of genetic databases and the improvement of enzyme properties via directed evolution should enable the design of artificial pathways and microbial cell factories for scalable production of 24-epi-ergosterol.

[1]Key Laboratory of Biomass Chemical Engineering of Ministry of Education, College of Chemical and Biological Engineering, Zhejiang University, Hangzhou 310027, China. [2]ZJU-Hangzhou Global Scientific and Technological Innovation Center, Zhejiang University, Hangzhou 310000, China. [3]Zhejiang Key Laboratory of Smart Biomaterials, Zhejiang University, Hangzhou 310027, China. ✉e-mail: linjp@zju.edu.cn; jzlian@zju.edu.cn

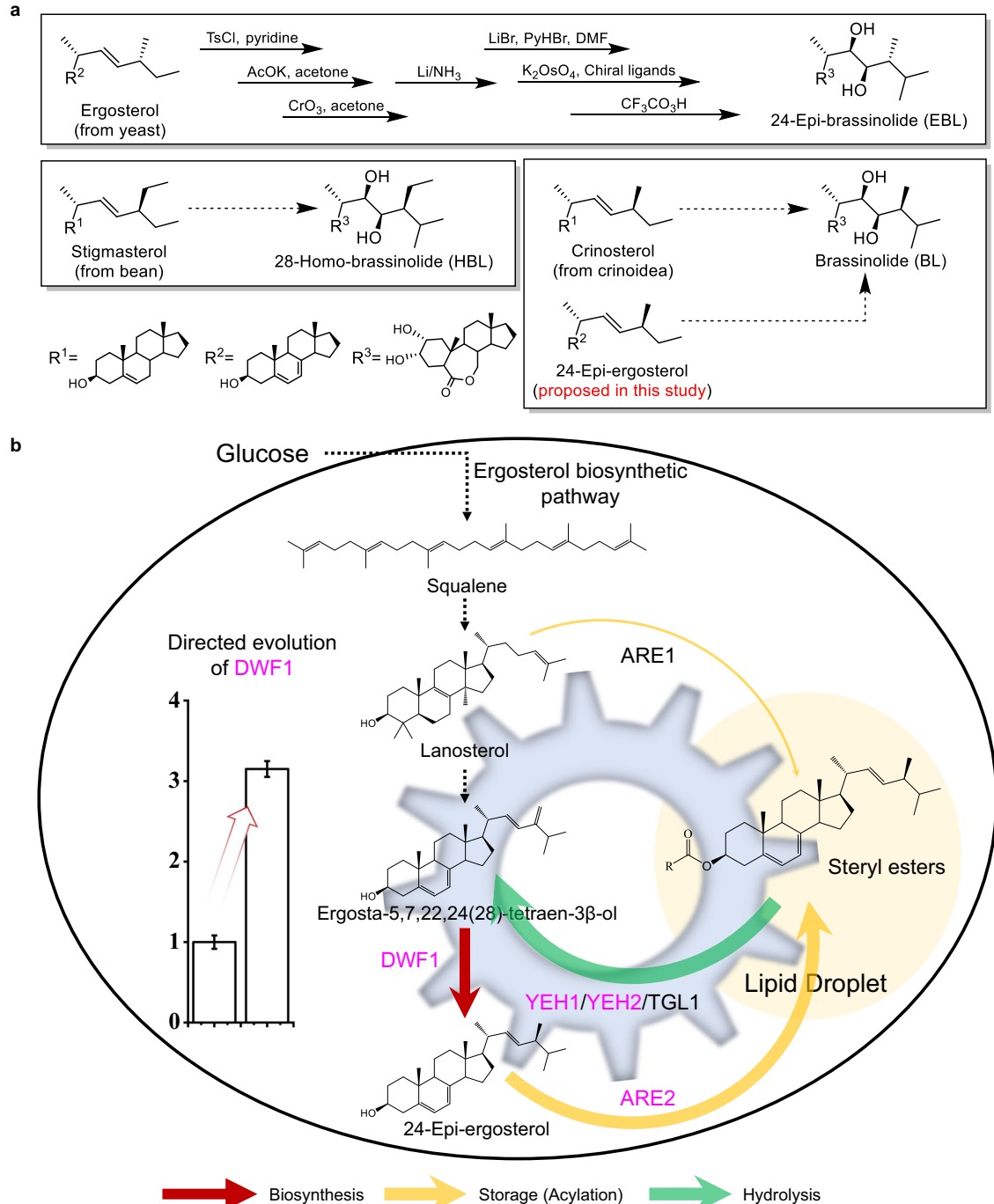

**Fig. 1 | Construction of engineered yeast strains for de novo biosynthesis of 24-epi-ergosterol, serving as a synthetic precursor for BL. a** Semi-synthesis of 24-epi-brassinolide (EBL), 28-homo-brassinolide (HBL), and brassinolide (BL) from ergosterol, stigmasterol, and crinosterol, respectively. In comparison with crinosterol, 24-epi-ergosterol, the diastereoisomer to ergosterol, could be produced on a large scale using yeast strains constructed in the present study. **b** Manipulation of sterol homeostasis for the production of 24-epi-ergosterol in yeast. The introduction of a $\Delta^{24(28)}$-sterol reductase (DWF1) from plants enabled de novo biosynthesis of 24-epi-ergosterol. Afterward, the catalytic activity of DWF1 and, accordingly, the production of 24-epi-ergosterol was enhanced by directed evolution. The sterol fluxes towards 24-epi-ergosterol were further strengthened by the engineering of sterol homeostasis, maintaining a balance between sterol acylation (for storage in LDs) and steryl ester hydrolysis (for releasing free sterols in cellular membranes) via overexpression of *YEH1*, *YEH2*, and *ARE2* with intact *ARE1*. DWF1 $\Delta^{24(28)}$-sterol reductase, ARE1 sterol *O*-acyltransferase 1, ARE2 sterol *O*-acyltransferase 2, YEH1 yeast steryl ester hydrolase 1, YEH2 yeast steryl ester hydrolase 2, TGL1 triglyceride lipase.

In addition to the lack of 24-epi-ergosterol biosynthetic pathway, the tight regulation of sterol metabolism represents another major challenge to achieve high-level production. The sterol biosynthetic pathway in *Saccharomyces cerevisiae* has been studied in depth, which is rather complex, with almost 30 enzymes involved. In addition, ergosterol biosynthesis is subject to feedback regulation at transcriptional, translational, and posttranslational levels to strictly control the intracellular sterol contents[10]. Furthermore, to alleviate cytotoxicity, excessive free sterols, including both upstream sterols and downstream sterols, are acylated and stored in lipid droplets (LDs)[11]. The sterol homeostasis between LDs and cellular membranes depends on two ER-localized acyl-CoA:sterol acyltransferases (ARE1

and ARE2)[12] and three steryl ester hydrolases (YEH1, YEH2, and TGL1)[13,14]. Previous metabolic engineering efforts for the construction of sterol-overproducing yeast strains generally focus on boosting the metabolic fluxes towards sterol biosynthesis[15]. For example, overexpression of *UPC2* (particularly the dominant mutant *UPC2-1*)[16,17] or *ECM22*[18], two zinc-cluster transcription factors, upregulated the expression of multiple ergosterol biosynthetic pathway genes (e.g., *ERG1*, *ERG11*, *ERG25*, *ERG6*, *ERG2*, and *ERG3*). As several of the ergosterol biosynthesis enzymes are localized in the endoplasmic reticulum (ER)[19], ER expansion has been employed to increase the yield of protopanaxadiol[20], medicagenic acid, and ergosterol[21]. On the other hand, sterols are mainly stored in LDs, and increasing the number and volume of LDs have been found to enhance steroid accumulation and alleviate adverse effects on growth[22]. What's more, sterol production could be enhanced by re-localizing ER-targeted proteins to LDs to eliminate the spatial separation between enzymes and substrates[23]. Recently, overexpression of *ARE2* and *ACC1* (encoding acetyl-CoA carboxylase) was found to be optimal for the accumulation and production of ergosterol[24]. In other words, sterol homeostasis should be carefully manipulated and the biosynthesis, storage, and hydrolysis of sterols should be balanced for the production of 24-epi-ergosterol in yeast.

In this work, we aim to establish a yeast cell factory for scalable production of 24-epi-ergosterol, serving as the precursor for BL synthesis (Fig. 1b). First, we design an artificial pathway for de novo biosynthesis of 24-epi-ergosterol by introducing $\Delta^{24(28)}$ sterol reductases (DWF1) from plants. Using a high-throughput screening method based on the hypersensitivity of the *erg4Δ* yeast to sodium dodecyl sulfate (SDS) and Hygromycin B (HygB), we perform directed evolution of DWF1 to increase 24-epi-ergosterol conversion efficiency. Subsequently, we engineer sterol homeostasis to boost metabolic fluxes towards 24-epi-ergosterol. The optimal yeast strains are constructed by overexpressing *ARE2*, *YEH1*, and *YEH2*, without disrupting *ARE1*, whose titer is further increased by enhancing the expression of *DWF1*, *ERG5* (encoding C-22 sterol desaturase), and *ACC1*. Finally, high-density cell fermentation enables the production of 24-epi-ergosterol with a titer of 2.76 g L$^{-1}$ and yield of 19.27 mg gDCW$^{-1}$. The present study highlights the significance of manipulating sterol homeostasis in achieving high-level production of 24-epi-ergosterol.

## Results

### Design of an artificial pathway for de novo production of 24-epi-ergosterol in yeast

As an unnatural sterol, no enzymes have been reported to convert ergosta-5,7,22,24(28)-tetraene-3β-ol to 24-epi-ergosterol directly, indicating the need to design an artificial biosynthetic pathway. Therefore, we chose to explore the genetic database of sterol biosynthesis in plants (Fig. 2a), with the chiral preference ($\Delta^{24(28)}$ reduction) and substrate structure similarity (sterol side chain) as the major criteria. By comparing sterol biosynthetic pathways in yeast and plants, we selected Dimunito/Dwarf1 (DWF1), a $\Delta^{24(28)}$ sterol reductase converting C28- and C29-$\Delta^{24(28)}$-olefinic sterols to 24-methyl- and 24-ethylcholesterols[25], as the best candidate for 24-epi-ergosterol production.

To test the production of 24-epi-ergosterol, *ERG4* was inactivated in S1-G-GU, which was based on an industrial yeast CICC1746 with the deletion of *GAL80* and overexpression of *UPC2-1*. The resultant yeast strain YQE102 was further engineered by introducing *DWF1* homologous genes from *Arabidopsis thaliana (AtDWF1)*[26], *Ajuga reptans (ArDWF1)*[25], *Brassica rapa (BrDWF1)*, and *Cannabis sativa (CsDWF1)* (listed in Supplementary Data 1), respectively (Supplementary Figs. 2, 3a). Luckily, the integration of the *DWF1* expression cassette into the *erg4Δ* yeast strain resulted in the formation of a sterol peak in our high-performance liquid chromatography (HPLC) analysis (Supplementary Fig. 3b), which was further confirmed to be

24-epi-ergosterol by mass spectrometry (MS, Supplementary Fig. 4) and nuclear magnetic resonance (NMR, Supplementary Fig. 5). Among multiple yeast strains, YQE224 bearing *ArDWF1* produced larger amount of 24-epi-ergosterol, with the titer reaching to 13.79 mg L$^{-1}$ (Supplementary Fig. 3a). Although the enzyme promiscuity of DWF1 enabled the production of 24-epi-ergosterol in yeast, the activity towards ergosta-5,7,22,24(28)-tetraen-3β-ol was still much lower than ERG4 and the production level was still far from industrial applications, indicating the necessity of further modifications by protein engineering and metabolic engineering.

### Directed evolution of DWF1 for the enhanced synthesis of 24-epi-ergosterol

After proof-of-concept verification of the biosynthesis of 24-epi-ergosterol, we next proposed to increase the enzymatic activity of DWF1 towards ergosta-5,7,22,24(28)-tetraen-3β-ol via protein engineering approaches. Due to the lack of crystal structures of DWF1 and homologous enzymes, we could only resort to directed evolution, for which a high-throughput screening (HTS) method is required. As previously reported, the *erg4Δ* yeast strain (accumulation of ergosta-5,7,22,24(28)-tetraen-3β-ol but not ergosterol as the final product) exhibited pleiotropic defects such as hypersensitivity to SDS[27] and HygB[28]. Considering the structural similarity, 24-epi-ergosterol should be able to complement ergosterol deficiency to some extent, based on which we could establish an HTS method by coupling DWF1 activity to cell growth under SDS and HygB stressed conditions. To facilitate library construction and HTS, the haploid *erg4Δ* strain YQE101 based on the laboratory yeast BY4741 was chosen as the chassis for directed evolution efforts.

To establish the growth-associated HTS method, we constructed and compared the growth rate of four yeast strains, YQP1 (BY4741 harboring pRS42H, high-level production of ergosterol, positive control), YQP2 (YQE101 harboring pRS42H, no ergosterol or 24-epi-ergosterol production, negative control), YQP3 (YQE101 harboring pRS42H-AtDWF1, low-level production of 24-epi-ergosterol), and YQP4 (YQE101 harboring pRS42H-ArDWF1, slightly higher production of 24-epi-ergosterol), in YPD medium supplemented with 100 µg mL$^{-1}$ HygB and different concentration of SDS. As shown in Fig. 2b, the yeast strain with *ERG4* (YQP1) grew normally with 0.01% SDS; the deletion of *ERG4* (YQP2) resulted in significant growth inhibition with 0.005% SDS; while the introduction of *DWF1* (YQP3 and YQP4), which enabled *erg4Δ* yeast to synthesize 24-epi-ergosterol, alleviated the growth inhibition with 0.01% and 0.005% SDS significantly. More importantly, the yeast strain (YQP4) producing a higher level of 24-epi-ergosterol exhibited better growth than a low-producing strain (YQP3), indicating a positive relationship between DWF1 activity (24-epi-ergosterol content) and cell growth under SDS and HygB stressed conditions.

We employed the established HTS method to screen the ArDWF1 mutant library generated by error-prone PCR (Supplementary Fig. 6a). About 2,000 colonies were selected from YPD/HygB + 0.01% SDS agar plates and tested for growth in YPD/Hyg + 0.025% SDS medium. Fortunately, we obtained six mutants (Supplementary Fig. 6b and Supplementary Table 2) with elevated growth rates and accordingly increased ArDWF1 catalytic activity. To evaluate the contribution of each mutation, we constructed 10 single mutants by site-directed mutagenesis and compared their performance (Fig. 2c). As all the mutations were found to be beneficial for ArDWF1 activity and it was impossible to test all the combinations individually, we performed multiplex-PCR-based recombination (MUPREC)[29] to construct a combinatorial library. Using the same HTS method, we obtained 28 positive combinatorial mutants, enabling better growth in YPD/Hyg + 0.025% SDS medium and higher ArDWF1 catalytic activity. The best mutant Ar207 with a combinatorial mutation of V143G, S306P, and Y338H showed higher ArDWF1 activity and 24-epi-ergosterol production,

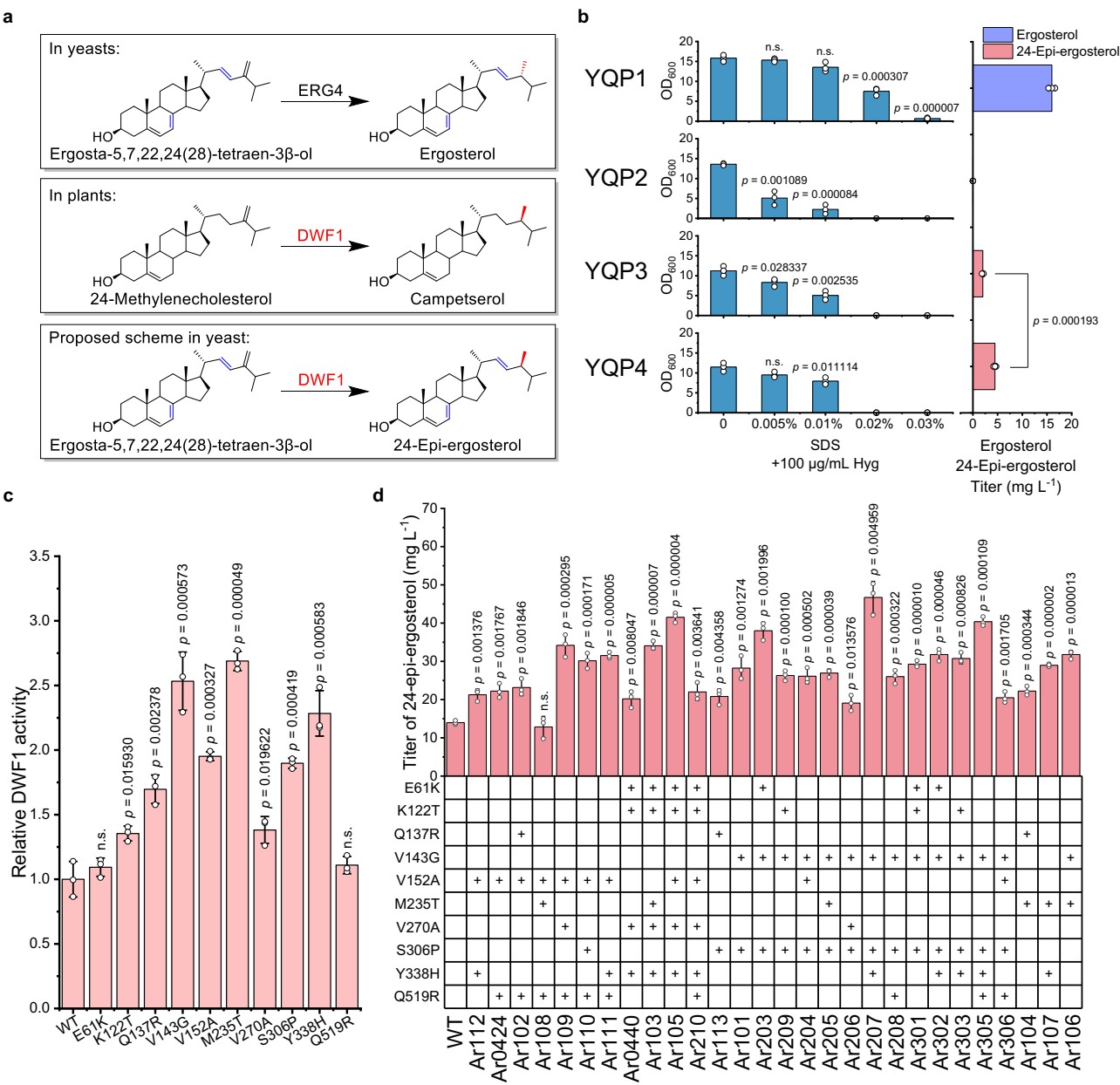

**Fig. 2 | Design and optimization of an artificial pathway for de novo biosynthesis of 24-epi-ergosterol. a** De novo biosynthesis of 24-epi-ergosterol using an artificial pathway. In yeasts, ergosta-5,7,22,24(28)-tetraene-3β-ol was converted to ergosterol by ERG4; In plants, 24-methylenecholesterol was catalyzed by DWF1 to synthesize campesterol. Based on the chiral preference and structural similarity, 24-epi-ergosterol was expected to be produced in yeast by introducing DWF1. ERG4, Δ24(28)-sterol reductase in yeasts; DWF1, Δ24(28)-sterol reductase in plants. **b** Development of an HTS method for directed evolution of DWF1. YQP1 (BY4741 harboring pRS42H), YQP2 (YQE101 harboring pRS42H), YQP3 (YQE101 harboring pRS42H-AtDWF1), YQP4 (YQE101 harboring pRS42H-ArDWF1) were constructed to examine the relationship between the growth under SDS with HygB stressed conditions and the ability to synthesize ergosterol (YQP1) or 24-epi-ergosterol (YQP3 and YQP4). SDS with different concentrations was used to screen higher 24-epi-ergosterol producing strains with positive DWF1 mutants. All yeast strains were cultivated in YPD medium at 30 °C for 72 h. SDS sodium dodecyl sulfate; Hyg

hygromycin. **c** The activity of DWF1 single mutants. Six mutants with a total of ten mutations (Supplementary Table 2) were obtained in the first round of directed evolution. The corresponding single mutants were constructed by site-directed mutagenesis and transformed into YQE101 as episomal plasmids for activity assays. To exclude the effect of HygB on the growth of ergΔ yeast strains (YQE101), the ratio of 24-epi-ergosterol over ergosta-5,7,22,24(28)-tetraene-3β-ol was employed to represent the enzymatic activity of DWF1 mutants. WT wild-type. **d** Activity of DWF1 combinatorial mutants. The second round of directed evolution via DNA shuffling resulted in the construction of a series of combinatorial mutants, whose activity was evaluated by genome integration into YQE102 (for 24-epi-ergosterol production). **b**–**d** Data were presented as mean values ± SD from three independent biological replicates (n = 3), the circles represent individual data points. Significance (p value) was evaluated by a two-sided t-test, no significance (n.s.) presents p > 0.05. Source data are provided as a Source Data file.

without losing stereoselectivity (Supplementary Fig. 4b). The introduction of Ar207 into YQE102 (YQE231) resulted in producing 24-epi-ergosterol to 46.72 mg L⁻¹, which was 3.46-fold higher than that of the control strain YQE224 (Fig. 2d and Supplementary Fig. 7).

## Manipulation of sterol homeostasis for enhanced 24-epi-ergosterol production

The sterol homeostasis between LDs and cellular membranes is mainly regulated by two sterol acyltransferases (ARE1 and ARE2)[12] and three

steryl ester hydrolases (YEH1, YEH2, and TGL1)[13,14]. While YEH1, YEH2, and TGL1 were able to hydrolyze all steryl esters, ARE1 mainly acylates early sterols (e.g., lanosterol) and ARE2 mainly contributed to the acylation of late sterols (sterols harboring 5,7-dien, e.g., ergosterol)[30]. To evaluate their contribution to sterol metabolism, we performed deletion or overexpression of each of these five genes, *ARE1*, *ARE2*, *YEH1*, *YEH2*, and *TGL1*. Although the disruption of *ARE1* or overexpression of *ARE2*, *YEH1*, and *YEH2* was beneficial for 24-epi-ergosterol biosynthesis, the improvement by single gene manipulation was rather limited, indicating a need to reconstruct sterol homeostasis through the engineering of sterol acylation and hydrolysis simultaneously (Fig. 3a). As expected, we observed a synergistic effect by simultaneous overexpression of acyltransferase genes (*ARE2*) and hydrolase genes (*YEH1* and/or *YEH2*). More specifically, strain YQE717 overexpressing *ARE2*, *YEH1*, and *YEH2* produced 71.04 mg L$^{-1}$ 24-epi-ergosterol and 220.07 mg L$^{-1}$ late sterols, which was 1.65- and 2.09-fold higher than that of YQE231, respectively (Fig. 3b). Notably, the introduction of an extra copy of *YEH1* or *YEH2* expression cassette failed to further increase the production of 24-epi-ergosterol or total late sterols.

Although the inactivation of *ARE1* (YQE722) resulted in comparable production of 24-epi-ergosterol in shake flasks (Fig. 3b and Supplementary Fig. 8), its performance in the fed-batch bioreactor was adversely affected, including cell growth and product formation (Fig. 3c and Supplementary Figs. 9, 10), particularly in the ethanol fermentation stage. After 120 h fermentation, the cell density of YQE717 reached 186.05 g dry cell weight per liter (gDCW L$^{-1}$), which was 153.36 gDCW L$^{-1}$ for YQE722. As a result, higher-level production of 24-epi-ergosterol was achieved in YQE717, with the titer and yield reaching to 2.15 g L$^{-1}$ and 11.53 mg gDCW$^{-1}$, respectively. To verify the contribution of sterol homeostasis engineering to the increased production of 24-epi-ergosterol, we analyzed the percentage of un-acylated sterols in YQE231 (un-modified except for the introduction of an ArDWF1 mutant), YQE710 (*ARE2*), YQE711 (*YEH1*), YQE712 (*YEH2*), YQE717 (*ARE2-YEH1-YEH2*), and YQE722 (*are1Δ-ARE2-YEH1-YEH2*). The overexpression of *ARE2, YEH1* or *YEH2* individually either increased or decreased the percentage of un-acylated sterols. On the other hand, simultaneous overexpression of *ARE2, YEH1*, or *YEH2* maintained the acylation ratio comparable to the un-modified yeast strain, indicating the reconstructed balance of sterol acylation and hydrolysis, which was disturbed by further inactivation of *ARE1* (Supplementary Fig. 11). The above results highlighted the significance of sterol homeostasis, which should be carefully manipulated for the production of 24-epi-ergosterol and other sterol compounds.

In addition, we performed quantitative PCR (qPCR) to determine the relative transcriptional level of *ARE2, YEH1*, and *YEH2* in YQE231 and YQE717 (Supplementary Fig. 12). In the first 36 h, the production of 24-epi-ergosterol and total sterols were comparable (Supplementary Fig. 8). Afterwards, sterol biosynthesis was nearly halted in YQE231, while YQE717 continued to accumulate significant amounts of sterols and 24-epi-ergosterol. The product formation pattern agreed well with the expression of *ARE2*, *YEH1*, and *YEH2*, which showed higher expression during 24–48 h in YQE717. These results suggested that the increased production of 24-epi-ergosterol was attributed to the enhanced transcriptional level of *ARE2*, *YEH1*, and *YEH2*, or the reconstructed sterol homeostasis in yeast. Considering the accumulation of sterols during the ethanol fermentation stage, the expression of *ARE2*, *YEH1*, and *YEH2* driven by ethanol inducible promoters might result in increased production of 24-epi-ergosterol. Therefore, *ARE2*, *YEH1*, and *YEH2* driven by different promoter combinations were investigated (Supplementary Table 3)[31]. Compared with YQE717, YQE717 SES, and YQE717 WSW resulted in a slightly higher level of production of 24-epi-ergosterol, whose total sterol levels were unfortunately decreased dramatically (Supplementary Fig. 13). Therefore, YQE717 was chosen for further metabolic engineering studies.

## Increased production of 24-epi-ergosterol by strengthening pathway gene expression

Although the production of 24-epi-ergosterol was significantly increased, ergosta-5,7,22,24(28)-tetraen-3β-ol, the substrate for DWF1, was still accumulated at high levels (Fig. 3b). The accumulation of ergosta-5,7,22,24(28)-tetraen-3β-ol not only lowered the production of the desired product, but also increased downstream purification cost. To achieve higher ratio production of 24-epi-ergosterol, we evaluated several promoters (i.e. $P_{TEF1}$, $P_{TDH3}$, $P_{ERG4}$, $P_{ERG5}$, $P_{CIT2}$, and $P_{GAL1}$) to drive the expression of ArDWF1 mutant (Ar207). When $P_{CIT2}$ and $P_{GAL1}$ were used, the production of 24-epi-ergosterol was dramatically increased, representing 1.44- and 1.97-fold higher than that with the original $P_{TEF1}$, respectively (Fig. 4a). In addition, the overexpression of *ACC1*, encoding acetyl-CoA carboxylase, has been reported to strengthen fatty acids biosynthesis and accordingly expand 24-epi-ergosterol storage pool[24]. Thus, we replaced the endogenous $P_{ACC1}$ with $P_{GAL10}$ to increase the expression level of *ACC1* (Supplementary Fig. 14). The resultant strain (YQE729) further increased the production of 24-epi-ergosterol to 160.84 mg L$^{-1}$ (Fig. 4a). Interestingly, the expression of *Ar207* driven by $P_{GAL1}$ caused the accumulation of 24-epi-ergosta-5,7-dien-3β-ol, another precursor in the 24-epi-ergosterol biosynthetic pathway. Thus, we constructed YQE734 by overexpressing *ERG5* (Supplementary Fig. 14), encoding the Δ$^{22}$ sterol desaturase, which decreased the accumulation of 24-epi-ergosta-5,7-dien-3β-ol from 43.02 to 3.27 mg L$^{-1}$ and accordingly increased the production of 24-epi-ergosterol from 160.84 to 171.31 mg L$^{-1}$, without the production of any ergosterol (Supplementary Fig. 15).

Finally, we evaluated the performance of YQE729 and YQE734 in 2 L fed-batch bioreactors (Supplementary Figs. 16, 17). Due to the enhanced expression of *Ar207* driven by $P_{GAL1}$ and overexpression of *ERG5*, ergosta-5,7,22,24(28)-tetraen-3β-ol and 24-epi-ergosta-5,7-dien-3β-ol were not accumulated to high levels, with the ratio of 24-epi-ergosterol to total late sterols reaching to 84.2% in YQE734. On the contrary, 1.68 g L$^{-1}$ 24-epi-ergosta-5,7-dien-3β-ol was accumulated at the end of fermentation in YQE729, with the ratio of 24-epi-ergosterol to total late sterols dropping to 52.1% (Fig. 4b). Notably, the total late sterols in YQE734 was lower than that in YQE729, probably due to metabolic burden caused by the overexpression of *ERG5*. In other words, we should fine-tune the expression level of *ERG5* in our future studies. Overall, although YQE729 and YQE734 produced a similar amount of 24-epi-ergosta (~2.76 g L$^{-1}$), high ratio of the target product in total sterols made YQE734 more promising for industrial applications.

## Discussion

Due to the lack of biosynthetic pathways and the scarcity of synthetic precursors, Brassinolide (BL), the most active BR compound, has not been widely applied in agriculture and industry yet. In the present study, we constructed an artificial biosynthetic pathway, followed by protein engineering and metabolic engineering, for the production of 24-epi-ergosterol in yeast, which could serve as a scalable route for the semi-synthesis of BL.

Although we have performed two rounds of directed evolution (the first round by error-prone PCR and the second round by DNA shuffling), the activity of ArDWF1 is still rather low, particularly when compared with that of ERG4 towards ergosta-5,7,22,24(28)-tetraen-3β-ol. In addition, the lack of crystal structures and the molecular mechanisms for improved enzymatic activity make further engineering of ArDWF1 rather challenging. To explore the possible mechanisms and provide insights for further protein engineering, we employ the recently developed Alpha fold2 model[32] to predict the 3D structure of ArDWF1. As a number of the VAO/PCMH (vanillyl alcohol oxidase/para-cresol methylhydroxylase) flavoprotein family, DWF1 shared a conserved FAD-binding domain, where K122, Q137, V143, and V152 are located[33]. V143, V152, and Y338 are located in the interface between the

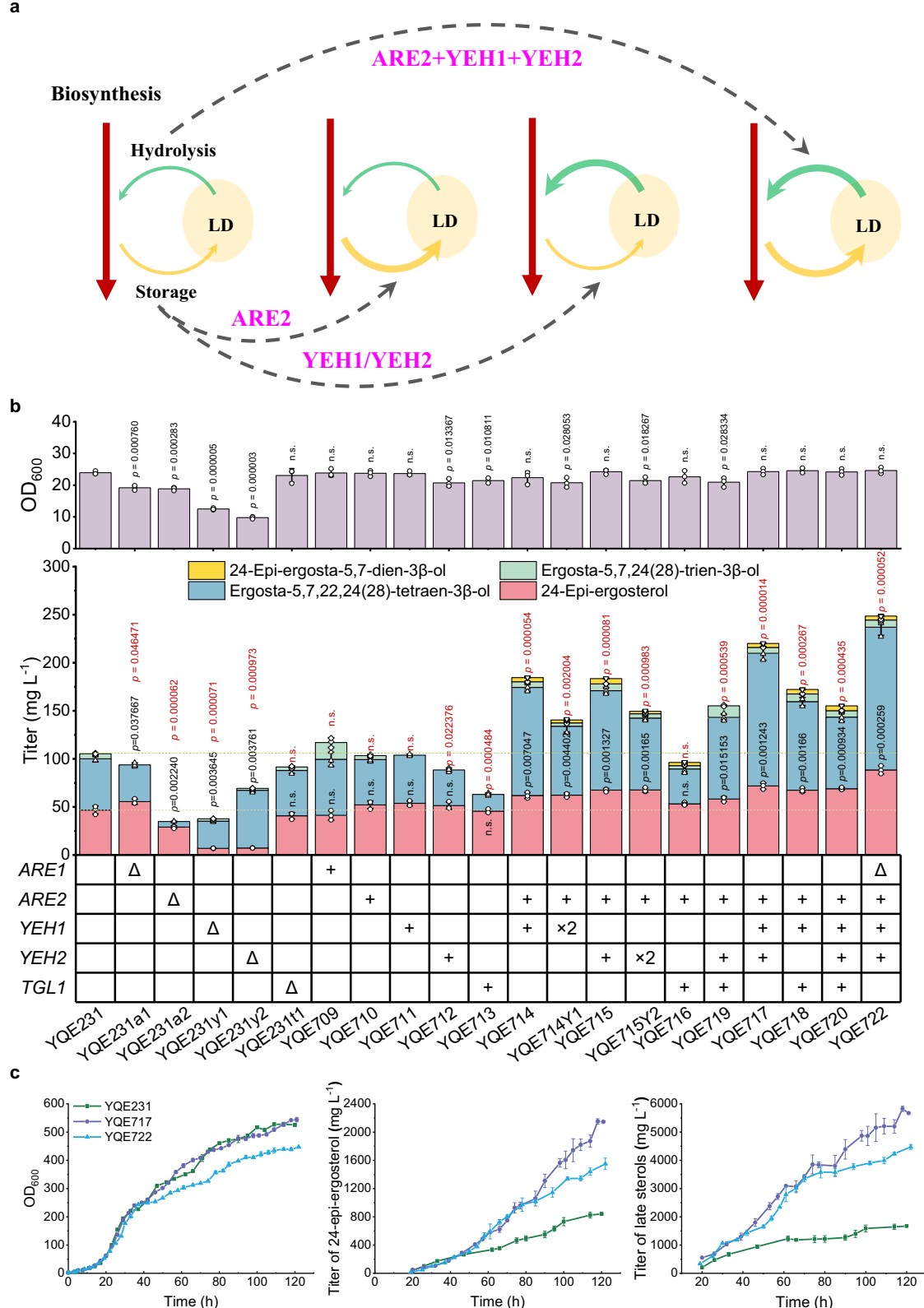

**Fig. 3 | Manipulation of sterol homeostasis for the production of 24-epi-ergosterol in yeast. a** Scheme of sterol homeostasis for maintaining a balance between sterol acylation (ARE1 and ARE2) and steryl ester hydrolysis (YEH1 and YEH2). LD lipid droplet. **b** The effect of sterol homeostasis engineering strategies on the production of 24-epi-ergosterol. The sterol acylation pathway (*ARE1* and *ARE2*) and steryl ester hydrolysis pathway (*YEH1*, *YEH2*, and *TGL1*) genes were overexpressed and/or knocked out, and their effects on the production of 24-epi-ergosterol and other late sterols were evaluated both individually and combinatorially. **c** Comparison of the fed-batch fermentation performance for YQE231,

YQE717, and YQE722. Fermentation profiles of cell growth, titer of 24-epi-ergosterol, and titer of late sterols (here defined as the total amount of ergosta-5,7,24(28)-trien-3β-ol, ergosta-5,7,22,24(28)-tetraen-3β-ol, ergosta-5,7-dien-3β-ol, 24-epi-ergosta-5,7-dien-3β-ol, ergosterol, and 24-epi-ergosterol) were obtained by analyzing samples every 4–8 h. **b, c** Data were presented as mean values ± SD from three independent biological replicates (*n* = 3). **b** The circles represent individual data points. Significance (*p* value) of the titer of total late sterols (red) and 24-epi-ergosterol (black) were evaluated by a two-sided *t*-test, no significance (n.s.) presents *p* > 0.05. Source data are provided as a Source Data file.

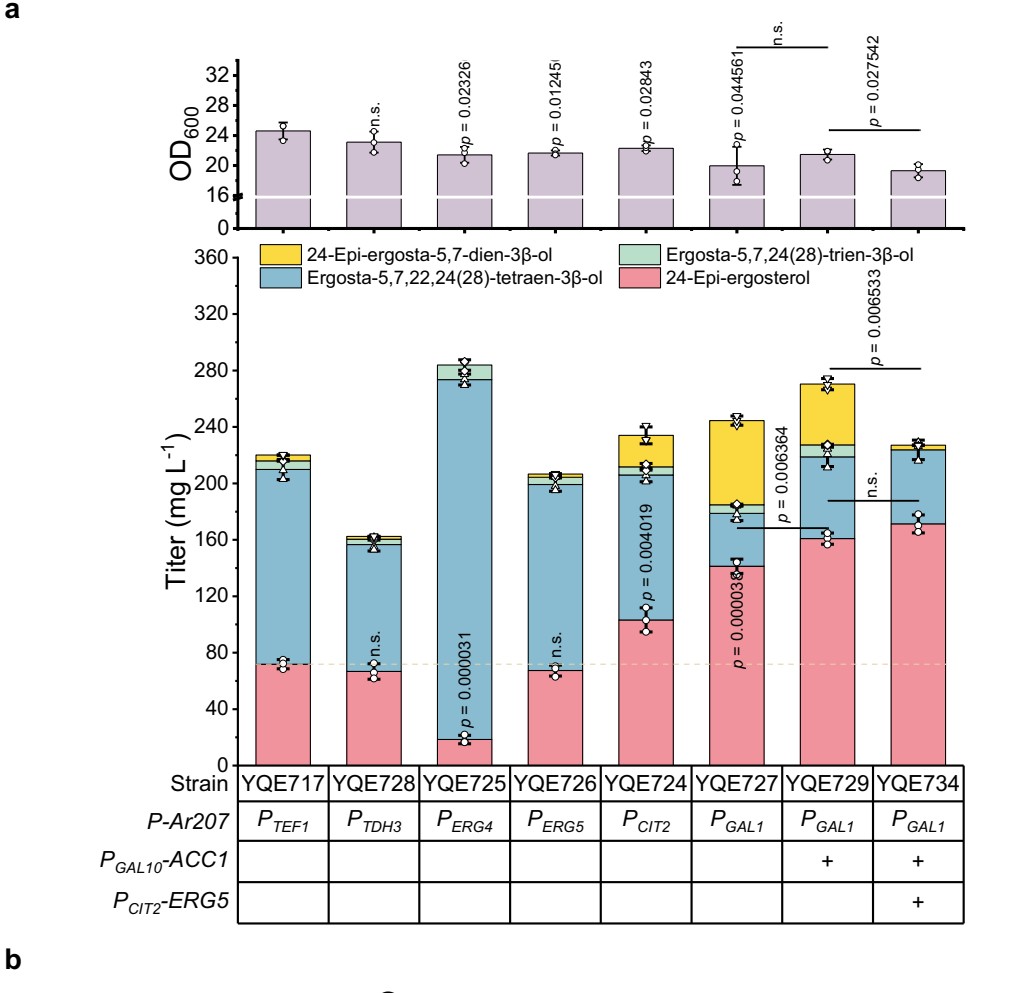

**Fig. 4 | Biosynthetic pathway engineering for the production of 24-epi-ergosterol. a** Enhancing 24-epi-ergosterol production by regulating the expression level of *Ar207*, *ACC1*, and *ERG5*. The expression level of *Ar207* was regulated by various promoters, including $P_{TEF1}$, $P_{TDH3}$, $P_{ERG4}$, $P_{ERG5}$, $P_{CIT2}$, and $P_{GAL1}$. The expression of *ACC1* and *ERG5* was enhanced by replacing the endogenous promoter with $P_{GAL10}$ and $P_{CIT2}$, respectively. *P-Ar207* the promoter chosen for *Ar207*, $P_{TEF1}$ the promoter of *TEF1* (encoding translation elongation factor 1), $P_{TDH3}$ the promoter of *TDH3* (encoding triose-phosphate dehydrogenase), $P_{ERG4}$ (encoding Δ$^{24(28)}$-sterol reductase), $P_{ERG5}$ the promoter of *ERG5* (encoding C-22

sterol desaturase), $P_{CIT2}$ the promoter of *CIT2* (encoding citrate synthase), $P_{GAL1}$ the promoter of *GAL1* (encoding galactokinase), ACC1 acetyl-CoA carboxylase, ERG5 C-22 sterol desaturase. **b** Fermentation profiles of YQE729 and YQE734 in fed-batch bioreactors. Time courses of cell growth, titer of 24-epi-ergosterol, and the ratio of 24-epi-ergosterol to late sterols were obtained by analyzing samples every 4–8 h. **a, b** Data were presented as mean values ± SD from three independent biological replicates ($n = 3$). **a** The circles represent individual data points. Significance ($p$ value) was evaluated by a two-sided $t$-test, no significance (n.s.) presents $p > 0.05$. Source data are provided as a Source Data file.

FAD-binding domain and the catalytic domain, and the substitution by smaller amino acid residues will make the domain more flexible to promote FAD turnover. In mutant Ar207 (combining V143G, S306P, and Y338H), the width of this interface is significantly increased (Supplementary Fig. 18). M235T is located at the periphery of the active domain, and the nucleophilic hydroxyl group may contribute to the formation of a hydrogen bond with 3β-OH, leading to improved ArDWF1 activity. The molecular mechanism for S306P, a site far from the catalytic center, is yet to be explored. Based on the predicted

structures, most positive mutation sites are located at the interface between the FAD-binding domain and catalytic domain, serving as the entry channel for a substrate. Substitution by smaller amino acid residues is expected to create more space for substrate entry. In other words, sterol reductase like DWF1 could be strengthened by engineering a more flexible interface between the FAD-binding domain and the catalytic domain.

As major components of LDs, steryl esters are formed by the acylation of free sterols via ARE1 and ARE2[34]. While ARE1 preferentially

acylates early sterol intermediates like lanosterol, ARE2 utilizes late sterols as the major substrates, especially ergosterol. Thus, deletion of *ARE1* is expected to lower precursor accumulation and push sterol fluxes towards late sterols, which can be further acylated by ARE2 for storage in LDs. Thus, most of the previous metabolic engineering efforts mainly focused on enhancing biosynthesis (such as over-expression of the ergosterol biosynthetic pathway genes) and strengthening storage (such as deletion of *ARE1* and/or overexpression of *ARE2*), while paid less attention to the reversed process, steryl ester hydrolysis (*YEH1, YEH2,* and *TGL1*). In the present study, we found that enhancing the hydrolysis of steryl ester could not only improve the sterol fluxes towards the final products, but also significantly increased the total amount of late sterols (Fig. 3b). More specifically, over-expression of *ARE2, YEH1,* and *YEH2* (YQE717 and YQE734) resulted in optimal performance of the engineered yeast strains for producing 24-epi-ergosterol. Interestingly, the inactivation of *are1* (YQE722) led to impaired cell growth and lower production of 24-epi-ergosterol (Fig. 3c). In *are1* deficient yeast strains, un-acylated lanosterol was accumulated to a higher level, which would stimulate DOA10 (ubi-quitin ligase complex)-dependent degradation of ERG1, the first and major rate-limiting step in the post-squalene pathway[30,35]. In other words, a circular metabolism may perform better than the linear counterpart for metabolic engineering applications. Notably, although the production of 24-epi-ergosterol and total late sterols were sig-nificantly increased in strain (YQE717) overexpressing *ARE2, YEH1,* and *YEH2* simultaneously (Fig. 2b), the ratio of un-acylated sterols was similar to that of the un-modified strain YQE231 (Supplementary Fig. 11). On the contrary, single-gene overexpression, as well as *are1* deletion, resulted in obvious changes in the ratio of un-acylated sterols (Supplementary Fig. 11), indicating the significance of maintaining a balance between sterol acylation and steryl ester hydrolysis for the production of 24-epi-ergosterol. Such a sterol homeostasis engineer-ing strategy can be applied to producing other sterol compounds in yeast.

In summary, we constructed an artificial pathway, followed by protein engineering and sterol homeostasis engineering, for the pro-duction of 24-epi-ergosterol in *S. cerevisiae*, promising as a synthetic precursor for scalable production of BL. Using fed-batch fermentation, we were able to produce 2.76 g L$^{-1}$ 24-epi-ergosterol, with the yield reaching 19.27 mg gDCW$^{-1}$.

# Methods

## Chemicals and reagents

All chemicals and kits were purchased from Sangon Biotech (Shanghai, China) unless specifically mentioned. The ergosterol standard, poly-ethylene glycol (PEG), and deoxyribonucleic acid sodium salt from salmon testes (single-stranded DNA, ssDNA) were purchased from Sigma-Aldrich. PrimeStar DNA polymerase, Ex Taq® DNA Polymerase, and all restriction enzymes were purchased from Takara (Dalian, China). Phanta® Max Super-Fidelity DNA Polymerase and ClonExpress II One Step Cloning Kit were purchased from Vazyme (Nanjing, China). Yeast Plasmid Extraction Kit and DNA Gel Purification Kit were pur-chased from Solarbio Life Science (Beijing, China) and Thermo Scien-tific (Massachusetts, USA), respectively.

## Strains and media

Yeast strains used in this study are listed in Supplementary Data 2. Yeast strain S1 (CICC1746) is an industrial strain for ergosterol pro-duction and was obtained from the China Center of Industrial Culture Collection. BY4741 is kindly provided by Prof. Zhinan Xu (Zhejiang University, China). CICC1746 genomic DNA was used for the amplifi-cation of *ARE1, ARE2, YEH1, YEH2,* and *TGL1*. *Escherichia coli* Trans-T1 (TransGen Biotech, China) was used as the host to construct, maintain, and amplify plasmids. *E. coli* strains were cultured in LB medium with

50 µg mL$^{-1}$ ampicillin. Yeast strains were cultivated in YPD medium (1% yeast extract, 2% peptone, and 2% D-glucose), with 100 µg mL$^{-1}$ hygromycin B (HygB) and 200 µg mL$^{-1}$ geneticin (G418) supplemented when necessary. YPD medium containing 100 µg mL$^{-1}$ HygB and 0.025% SDS was used for high-throughput screening of ArDWF1 mutants.

## Plasmid construction

All plasmids and primers (synthesized by Sangon Biotech, Shanghai, China) used in this study are listed in Supplementary Data 3 and Supplementary Data 4, respectively. Kits used in DNA manipulation were purchased from Sangon Biotech (Shanghai, China). General DNA amplification from genomic DNA was carried out according to the standard protocol of Phanta® Max Super-Fidelity DNA Polymerase (Vazyme, Nanjing, China). A yeast plasmid extraction kit was pur-chased from Solarbio Life Science (Beijing, China).

*DWF1* genes (*AtDWF1, ArDWF1, BrDWF1,* and *CsDWF1*) were codon-optimized for yeast expression and synthesized by Sangon Biotech, which were subsequently cloned into pRS42H, resulting in the con-struction of the plasmid pRS42H-AtDWF1, pRS42H-ArDWF1, pRS42H-BrDWF1, and pRS42H-CsDWF1, respectively. Plasmid pRS42H-SpCas9 and pKan100-ADE2.1[36], from Prof. Huimin Zhao (University of Illinois at Urbana-Champaign, Urbana, Illinois), were used for genome editing in yeast. Specific N20 (from E-CRISP online tool[37]) of the guide RNA (gRNA) was introduced in the primers used for amplification of the entire pKan100-ADE2.1 plasmid by inverse polymerase chain reaction (PCR)[38]. The PCR product was then digested by *Dpn*I and transformed into *E. coli* for amplification.

For co-expression of genes involved in sterol acylation and sterol ester hydrolysis, pEB-3-11 with three expression cassette was con-structed based on *pEASY*®-Blunt Simple Cloning Vectors (TransGen Biotech, Beijing, China). To replace $P_{TEF1}$ in pRS42H-Ar207 with $P_{TDH3}$, $P_{ERG4}$, $P_{ERG5}$ $P_{CIT2}$, and $P_{GAL10}$-$P_{GAL1}$, the vector scaffold was amplified by reverse PCR using R-pRS42H-207-F and R-pRS42H-207-R, the desired promoters were obtained by PCR from CICC1746 genome, which were recombined to construct pRS42H-$P_{TDH3}$-Ar207, pRS42H-$P_{ERG4}$-Ar207, pRS42H-$P_{ERG5}$-Ar207, pRS42H-$P_{CIT2}$-Ar207, and pRS42H-$P_{GAL10-PGAL1}$-Ar207, respectively.

As for the construction of donor DNAs for the deletion of *ERG4, ERG5, ARE1, ARE2, YEH1, YEH2,* and *TGL1*, the upstream and down-stream fragments were amplified from the CICC1746 and pieced together using overlap extension PCR. The full-length donor DNA fragments were gel purified and cloned into the *pEASY*®-Blunt Simple Cloning Vectors (TransGen Biotech, Beijing, China), to create pEASY-erg4Δ, pEASY-erg5Δ, pEASY-are1Δ, pEASY-are2Δ, pEASY-yeh1Δ, pEASY-yeh2Δ, and pEASY-tgl1Δ, respectively. To construct donor DNAs for the integration of *ARE1, ARE2, YEH1, YEH2,* and *TGL1* expression cassettes, the corresponding genes were cloned into pRS42H using ClonExpress II One Step Cloning Kit (Vazyme, Nanjing, China).

## Yeast transformation and strain construction

CRISPR/Cas9 guided gene knockout and integration were performed with some modifications[36]. Yeast cells were transformed by the PEG/ssDNA/LiAc method[39]. The Cas9-expressing strains were constructed by transforming pRS42H-SpCas9 (harboring *HygB* for hygromycin B resistance) into the corresponding yeast strains. For the co-transformation of gRNA expression plasmids (harboring *KanMX* for G418 resistance) and donor DNAs into Cas9-expressing strains, heat shock time was prolonged to 50 min, and the yeast strains were recovered in 1 mL YPD for 6 h to allow sufficient expression of the G418 resistance gene. Then the positive transformants harboring pRS42H-SpCas9 and gRNA expression plasmid were selected on YPD/HygB +G418 plates and subsequently confirmed by colony PCR and DNA sequencing.

## Directed evolution of DWF1

For the directed evolution of DWF1, the mutant library with two 35 bp homologous arms to $P_{TEF1}$ and $T_{ADH2}$ in pRS42H was generated by error-prone PCR using Ex Taq® DNA Polymerase (Takara, Dalian, China) with primer pairs of Ep-ArDWF1-F & Ep-ArDWF1-R. The concentration of $Mn^{2+}$ was set as 0.07 mM. Recombination-mediated PCR-directed plasmid construction in vivo in yeast was performed with some modifications[40]. Linearized vector and insert fragments from error-prone PCR were co-transformed into YQE101 in a molar ratio of 1:6, and then the transformants were selected on YPD/HygB+0.01% SDS plates. The colonies in the plates were picked and cultured in 2 mL YPD medium containing 100 µg mL$^{-1}$ HygB as well as 0.025% SDS in 24-well plates. After three days, the grown strains were selected and cultured in 5 mL YPD/HygB to quantify the production of 24-epi-ergosterol. To minimize the effect of HygB on cell growth, the activity of DWF1 was defined as the ratio of the HPLC peak areas of 24-epi-ergosterol and ergosta-5,7,22,24(28)-tetraen-3β-ol. Plasmids of the positive mutants were extracted using a yeast plasmid extraction kit (Solarbio, Beijing, China) and transformed into E. coli. The corresponding mutations were verified by DNA sequencing. To evaluate the contribution of each mutation to improved DWF1 activity, single mutants of DWF1 were constructed by reverse PCR-based site-directed mutagenesis using PrimeStar DNA polymerase (Takara, Dalian, China).

## 24-epi-ergosterol production in shake flasks and fed-batch bioreactors

Shake-flask experiments were carried out in biological triplicates in 250 mL shake flasks containing 50 mL YPD. Single colonies were inoculated into 5 mL YPD and incubated at 30 °C for 24 h, and then transferred to 250 mL shake flasks with an initial $OD_{600}$ of 0.1. The yeast strains were cultured at 30 °C and 220 rpm for 96 h.

For fed-batch cultivation, Single colonies were inoculated into 5 mL YPD medium and cultured at 30 °C and 220 rpm for 24 h and then transferred into 250 mL shake flasks containing 50 mL of YPD medium. After 20 h cultivation, yeast cells from two shake flasks were used to inoculate 0.9 L fermentation medium (10 g L$^{-1}$ D-glucose, 10 g L$^{-1}$ (NH4)$_2$SO$_4$, 8 g L$^{-1}$ KH$_2$PO$_4$, 3 g L$^{-1}$ MgSO$_4$, 0.72 g L$^{-1}$ ZnSO$_4$.7H$_2$O, 10 mL L$^{-1}$ trace metal solution, and 12 mL L$^{-1}$ vitamin solution) in a 2 L bioreactor (T&J-MiniBox, Shanghai, China). Fermentations were carried out at 30 °C, and pH was controlled at 5.0 by the automatic addition of 5 M ammonia hydroxide. Dissolved oxygen (DO) was maintained at >25% saturation by adjusting the agitation rate (300 to 950 rpm) and airflow rate (1 vvm to 3 vvm). After the complete consumption of initial glucose and residual ethanol, a feeding solution containing 500 g L$^{-1}$ glucose and 12 mL L$^{-1}$ vitamin solution was fed into the bioreactor, based on the pseudo-exponential feeding model developed by O'Connor[41]. Afterward, to improve the intracellular accumulation of 24-epi-ergosterol, ethanol was fed to the fermenter at a rate of 5 or 6 mL h$^{-1}$, maintaining DO above 25%. Until the end of the fermentation. The feeding rate $F_S$ during the pseudo-exponential feeding phase was determined by the following equations[42]:

$$F_{S} = \left( \frac{\mu}{Y_{\frac{X}{S}}} + m \right) \cdot \frac{X_0 V_0}{S} \cdot e^{\mu t} \tag{1}$$

Where $X_O$, $V_O$, and $S$ were the initial biomass density (gDCW L$^{-1}$), the initial culture volume (L), and the glucose concentration (g L$^{-1}$) in the feeding medium; $Y_{X/S}$ was the yield of the cell biomass on glucose (gDCW per g glucose); $\mu$ was the specific growth rate (h$^{-1}$); $m$ was the maintenance coefficient (g glucose gDCW$^{-1}$ h$^{-1}$), and $t$ was the time (h) after starting the feeding. A predetermined specific growth rate[24] of 0.12 h$^{-1}$ was used to avoid overflow metabolism. The values of $Y_{X/S}$ and $m$ were 0.5 and 0.05, respectively[43]. The feeding rate was adjusted every hour, according to the theoretical model.

## Real-time quantitative PCR analysis

Total RNA was isolated from yeast cells by TRIzol (Invitrogen) according to the manufacturer's instructions. The degradation of genomic DNA and reverse transcription were conducted by ReverTra Ace™ qPCR RT Master Mix (TOYOBO, Japan). Real-time Quantitative PCR was performed by 2×T5 Fast qPCR Mix (SYBR Green I) (TSINGKE, China) on LineGene 9600 Plus FQD-96A (Bioer Technology, China). The ACT1 gene (encoding actin) was used as the internal control for expression level normalization. The transcription level was analyzed using the $2^{-\Delta\Delta CT}$ method[44]. The primers for ACT1, ARE2, YEH1, YEH2, ACC1, Ar2O7, and ERG5 were synthesized by TSINGKE, and the corresponding sequences were listed in Supplementary Data 4.

## Analytical methods

Cell growth was monitored by measuring optical density at 600 nm ($OD_{600}$) using a spectrophotometer (721 G, INESA, Shanghai, China). Glucose and ethanol concentrations were determined using a biosensor (SBA-40C; Biology Institution of Shandong Academy of Science, Jinan, China). 24-Epi-ergosterol and precursor sterols were extracted from yeast cells with some modification[45]. About 500 µL yeast cell culture was harvested and washed twice using ddH$_2$O. About 600 µL alcoholic KOH solution (25% [w/v] in 50% ethanol) was added to the yeast pellets, which were vortexed for 1 min. Cell suspensions were then boiled for 1 h. After cooling on ice, sterols were extracted with 800 µL petroleum ether, followed by a vigorous vortex for 3 min. About 500 µL petroleum ether (top) layer was collected and dried with a vacuum dryer. Dried samples were dissolved in 500 µL ethanol and analyzed by HPLC, equipped with a Thermo C-18 column (ODS Hypersil, 4.6 × 250 mm, 5 µm) and a UV detector at 280 nm. Methanol/acetonitrile (80:20, v/v) was used as the mobile phase with an elution rate of 1 mL min$^{-1}$. LC-MS was performed through AB Sciex Triple TOF 5600+, equipped with a Thermo C-18 column (ODS Hypersil, 4.6 × 250 mm, 5 µm) and an atmospheric pressure chemical ionization (APCI) ion source. Sterols were separated on methanol/acetonitrile (80:20, v/v) over 20 min with a flow rate of 0.8 mL min$^{-1}$ at 30 °C. MS was operated in positive ionization mode, with a scanned $m/z$ range of 100–1500. All results were reported as the average of at least three biological replicates.

The analysis of free sterols and sterol esters was performed with some minor modifications[46]. About 4 mL yeast cell culture was harvested, washed twice using ddH$_2$O, and resuspended in 800 µL TE buffer. Then, 1 g glass beads (425–600 µm, acid washed, Sigma) was used to disrupt the cell walls mechanically by vigorous vortex for 10 min (30 s vortex and 30 s on ice). Afterward, sterols were extracted with 3.2 mL petroleum ether, followed by vigorous vortex for 3 min. For each extraction, a 500 µL petroleum ether (top) layer was collected separately in two tubes and dried with a vacuum dryer. One tube was analyzed by HPLC directly to determine un-acylated late sterols, while the other was subject to a saponification procedure before quantitative analysis by HPLC to determine total late sterols.

## Preparation of 24-epi-ergosterol for NMR analysis

About 120 mL yeast cells were collected and resuspended in 240 mL ethanol−KOH solution (KOH: ethanol: H$_2$O = 25:50:50, w/v/v) in a 500 mL shake flask. The reaction mixtures were incubated at 95 °C for 2 h. After cooling to room temperature, the samples were extracted with petroleum ether (boiling point range, 60−90 °C) (3 × 200 mL). The organic layer was dried over Na$_2$SO$_4$ and evaporated, which was further purified by column chromatography using n-hexane:ethyl acetate (100:1) as the eluent. The resultant crude 24-epi-ergosterol sample also included the upstream precursors ergosta-5,7,22,24(28)-tetraen-3β-ol and ergosta-5,7-dien-3β-ol. The crude product was completely dissolved in 5−7 volumes of solvent (n-hexane:ethyl acetate = 1:1) at 60 °C and then cooled down to room temperature for 5 h. The precipitated solid was filtered and the recrystallization procedure was

performed for multiple cycles until the purity of 24-epi-ergosterol was sufficient for NMR analysis. The purified 24-epi-ergosterol and ergosterol standard were dissolved in $CDCl_3$ for 500 M NMR analysis (BRUKER DMX-500).

**Statistics and reproducibility.** All experimental data were at least in triplicate and expressed as mean ± standard error. All data analyses were performed by Excel or OriginPro.

### Reporting summary

Further information on research design is available in the Nature Portfolio Reporting Summary linked to this article.

## Data availability

Data supporting the findings of this work are available within the paper and its Supplementary Information files. A reporting summary for this Article is available as a Supplementary Information file. All the protein structures are obtained from Alpha fold2. Heterologous gene sequences, strains, plasmids, and primers used in this study are provided in Supplementary Data 1–4, respectively. Source data are provided with this paper.

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

## Acknowledgements

This work was supported by the National Key Research and Development Program (2019YFA0905404), the Natural Science Foundation of Zhejiang Province (LR20B060003), and the Natural Science Foundation of China (22278361). We thank Prof. Zhinan Xu (Zhejiang University, China) for sharing BY4741 and Prof. Huimin Zhao (University of Illinois at Urbana-Champaign, Urbana, Illinois) for generously providing pRS42H-SpCas9 and pKan100-ADE2.1.

## Author contributions

Y.J. and JP.L. designed the experiments. Y.J. performed the experiments, analyzed the data, and drafted the manuscript. Z.W. participated in the directed evolution of ArDWF1. Y.J., Z.S., and K.L. performed the bioreactor fermentation. Y.J., G.L., L.Z., and H.X. performed the establishment of quantitative analytical and separation methods for 24-epi-ergosterol. Y.J. and Y.F. performed qPCR analysis. M.W. and L.Y. participated in the discussion and coordination of the study. JZ.L. participated in data analysis, discussion, and revision of the manuscript. JP.L. supervised the whole research and revised the manuscript. All authors approved the manuscript.

## Competing interests

The authors declare no competing interests.
