## [Peer Review File · Nature Communications]

Manipulation of Sterol Homeostasis for High-level Production of 24-Epi-ergosterol in Industrial YeastReviewers' Comments:

Reviewer #1:

Remarks to the Author:

This work describes an artificial pathway for the biosynthesis of 24-epi-ergosterol in the industrial yeast. The authors use protein engineering and sterol homeostasis engineering to strengthen 24-epi-ergosterol synthesis and achieve 2.76 g/L production of 24-epi-ergosterol in fed-batch fermentation. In general, this article is well-written and interesting. Although protein engineering and metabolic engineering in *S.c* has been long established, the pathway, product, and titer are of sufficient novelty to merit publication in NC. However, there are some questions that should be addressed.

1. The authors proposed a strategy of enhancing 24-epi-ergosterol production through manipulating sterol homeostasis. However, whether the regulation is carried out by breaking the sterol homeostasis or maintaining the homeostasis needs to be explained based on the results.
2. Yeast strains S1 (CICC1746) is an industrial strain, the authors engineered the strain by multiple gene knockout and overexpression, how to screen the recombinants in this study?
3. L391, "The corresponding mutations were verified by DNA sequencing and re-introduced into the wild type DWF1 by reverse PCR based site-directed mutagenesis using PrimeStar DNA polymerase", what is the meaning?
4. In Supplementary Table S4, "pEB-3-11 pEASY; TADH2-BamHI-Hind III-PTEF1-PTDH3-XhoI-AflII-TCYC1- PFBA1-EcoRI-AvrII-TPGK1", why is the terminator ADH2 in front of the promoter TEF1?
5. In Figure 2B, YQP1 produced ergosterol, not 24-epi-ergosterol. The figure needs modification.
6. What's the meaning of "the batch culture phase" (L408) and "pseudo-exponential phase" (L413) ?
7. L420, "A predetermined specific growth rate of 0.12 h⁻¹", where was the number (0.12 h⁻¹) from? Why choose it?
8. There are some ambiguous abbreviations in the manuscript. For example: L169, "SDS stressed conditions"; L303, "DOA10"; L427, "gDNA"; L429, "ACT1". Please indicate full name when necessary.

Reviewer #2:

Remarks to the Author:

The manuscript of Jiang et al. describes the multiple manipulation of yeast strains for high-level production of 24-Epi-ergosterol, an un-natural sterol which can be used as precursor for the synthesis of the plant hormone Brassinolide. This phytohormone has great potential for agricultural and industrial applications, however, the natural abundance of Brassinolide is extremely low. Therefore, the generation of microbial cell factories for scalable production of 24-epi-ergosterol is of great importance.

The strategy to achieve this included several challenging approaches, starting with the design of an artificial pathway for de novo biosynthesis of 24-epi-ergosterol by introducing $\Delta 24(28)$ sterol reductases (DWF1) from plants. Directed evolution of DWF1 for enhanced synthesis of 24-epi-ergosterol and engineering the sterol homeostasis to boost metabolic fluxes towards 24-epi-ergosterol further increased the amount of the desired product. After evaluating several promoters, the yield of 24-epi-ergosterol synthesized by an optimized yeast strain could be increased to 2,76 gL⁻¹ by cultivation in a bioreactor. The results of this study are noteworthy and significant.

The manuscript is written in a clear manner, the figures are of very good quality.

However, there are some minor changes to be made before the manuscript can be accepted.

Minor changes:

Line 176: "The sterol homeostasis between LDs and cytoplasm...." - Esterified sterols are found in LDs and non-esterified sterol in cellular membranes - but not in the cytoplasm!

Sometimes the wording is wrong, e.g.:

Line 220: "...inducible promoters might resulted in.." - might result in

Line 266: "... which could be severed as..." - which could serve as

Some gene names are not written in italic - please correct this.

Figure 2B, right panel: As indicated, this panel shows the titer of 24-epi-ergosterol in different strains. However, according to the text in the manuscript (line 146 and 147), the strain YQP1 harboring the plasmid pRS42H is producing high levels of ergosterol and no 24-epi-ergosterol! This must be corrected in the Figure 2B.

Table S3: Please indicate what the colors in the table stand for.

Reviewer #3:

Remarks to the Author:

In this paper, the authors engineer an enzyme and yeast strain in order to produce 24-epi-ergosterol. To accomplish this the authors identified a plant enzyme with the needed enzymatic activity, further improved the production of that enzyme with directed evolution, and engineered the sterol homeostasis of the yeast strain to further improve product. Ergosterol has previously been used as the starting material for a semi-synthesis of 24-epi-brassinolide, a less active diastereomer of brassinolide. Therefore it is likely that using 24-epi-ergosterol as the starting material will make it possible to produce brassinolide. The increased production of 24-epi-brassinolide reported by the authors will enable those semi-synthetic studies.

The reported increases in expression are impressive, however I believe more data could be provided to further demonstrate that the desired product is being selectively produced.

My main concern is that the enzyme directed evolution may have reduced the stereoselectivity of the enzyme because there is no pressure in the selection to prevent the enzyme from losing stereoselectivity since production of either 24-epi-ergosterol and ergosterol should be sufficient for survival. The authors show HPLC and LC-MS data in the SI for the WT enzyme that nicely demonstrate that the WT enzyme is stereoselective and produces 24-epi-ergosterol and not ergosterol. But the same data are not shown for the final engineered enzyme Ar207. Therefore, it would be helpful to see the equivalent of Figure S3 and Figure S4 for the yeast strain containing Ar207 as well as the final best engineered strain YQE734 compared to an ergosterol standard to ensure that these strains are selectively producing 24-epi-ergosterol over ergosterol.

Another minor issue is that while all experiments were done in triplicate, there aren't any statistics reported for any of the results. In some cases the significance of differences reported are clear but other cases they are not. For example, some of the mutant enzymes (especially the single mutants E61K and Q519R) do not appear to be significantly better than WT but they are all reported as improvements in the text. I recommend adding indications of significance on the plots comparing titers or OD600 so it is easier to tell what differences are statistically significant, and mentioning p-values, confidence intervals, or some other measure of significance in the text when claiming significant improvements.

Besides from the two above mentioned issues, the methodology and interpretation of the data in the paper appears sound.

Reviewer 1

Comment 1: The authors proposed a strategy of enhancing 24-epi-ergosterol production through manipulating sterol homeostasis. However, whether the regulation is carried out by breaking the sterol homeostasis or maintaining the homeostasis needs to be explained based on the results.

Response: We appreciate the reviewer's valuable comments. Based on the two observations, 1) the high producing strain (YQE717 overexpressing *ARE2*, *YEH1*, and *YEH2* simultaneously) maintained the ratio of un-acylated sterols similar to that of the un-modified strain YQE231, indicating a sterol homeostasis status; 2) the low producing strains (single-gene overexpression as well as *are1* deletion) were found to change the ratio of un-acylated sterols significantly, indicating un-balanced sterol acylation and steryl ester hydrolysis; we think it necessary to maintain the sterol homeostasis to achieve high-level production of 24-epi-ergosterol and other phytosterols. We have briefly discussed this issue in Line 307-314 of the revised manuscript.

Comment 2: Yeast strains S1 (CICC1746) is an industrial strain, the authors engineered the strain by multiple gene knockout and overexpression, how to screen the recombinants in this study?

Response: Thanks for the reviewer's comments. The recombinants were constructed based on the CRISPR/Cas9 established for industrial yeast engineering in our previous studies. Basically, the Cas9 expressing plasmid pRS42H-SpCas9 harbors *HygB* for hygromycin B resistance and the gRNA expressing plasmid pKan100-SpSgH harbors *KanMX* for G418 resistance. After transformation using the PEG/ssDNA/LiAc method, recombinant yeast strains were selected on YPD/HygB+G418 plates and subsequently confirmed by colony PCR and DNA sequencing. To address the reviewer's concern, we have described the CRISPR/Cas9 system for industrial yeast engineering and the selection of recombinant yeast strains in more details in Line 376, Line 377, and Line 380-381 of the revised manuscript.

Comment 3: L391, “The corresponding mutations were verified by DNA sequencing and re-introduced into the wild type DWF1 by reverse PCR based site-directed mutagenesis using PrimeStar DNA polymerase”, what is the meaning?

Response: Thanks for the reviewer’s comments. Mutants obtained from the first round of directed evolution contained multiple mutations. To evaluate the contribution of each mutation to improved DWF1 activity, we constructed single mutants of DWF1 by site-directed mutagenesis. To make it easier to understand, we have revised the sentence as “The corresponding mutations were verified by DNA sequencing. To evaluate the contribution of each mutation to improved DWF1 activity, single mutants of DWF1 were constructed by reverse PCR based site-directed mutagenesis using PrimeStar DNA polymerase” (Line 398-402).

Comment 4: In Supplementary Table S4, “pEB-3-11 pEASY; *T_{ADH2}*-BamHI-HindIII-*P_{TEF1}*-*P_{TDH3}*-XhoI-AflIII-*T_{CYC1}*-*P_{FBA1}*-EcoRI-AvrII-*T_{PGK1}*”, why is the terminator *ADH2* in front of the promoter *TEF1*?

Response: Thanks for the reviewer’s comments. Due to the use of bi-directional promoters (*P_{TEF1}*-*P_{TDH3}*), the two expression cassettes are arranged in opposite directions. Following is the plasmid map for pEB-3-11.

Comment 5: In Figure 2B, YQP1 produced ergosterol, not 24-epi-ergosterol. The figure needs modification.

Response: Thanks for the reviewer's comments. Figure 2B has been revised to show ergosterol and 24-epi-ergosterol correctly (Figure 2B of the revised manuscript).

Comment 6: What's the meaning of "the batch culture phase" (L408) and "pseudo-exponential phase" (L413)?

Response: Thanks for the reviewer's comments and sorry for the misuse of "the batch culture phase". To minimize confusion, we have changed "the batch culture phase" to "After complete consumption of initial glucose and residual ethanol" in Line 417-418 of the revised manuscript.

As for "pseudo-exponential phase", it means a phase in three-phase fermentation strategy to achieve high-density fermentation. As the bioreactor control system cannot realize feeding rate in the exponential way, the feeding rate F_s was calculated in every hour to fit exponential feeding rate, the so-called "pseudo-exponential". To elucidate the process more clearly, we have changed the "pseudo-exponential phase" to "the pseudo-exponential feeding phase" and provided the corresponding reference in Line 422-423 of the revised manuscript.

Comment 7: L420, "A predetermined specific growth rate of 0.12 h^{-1} ", where was the number (0.12 h^{-1}) from? Why choose it?

Response: Thanks for the reviewer's comments. Fermentation conditions were optimized and high-density fermentation was achieved for CICC1746 in our lab previously¹. Therefore, the corresponding specific growth rate of 0.12 h^{-1} was chosen for the present study. To make it easier to follow, we have added a reference describing the specific number (Line 429 of the revised manuscript).

Comment 8: There are some ambiguous abbreviations in the manuscript. For example: L169, "SDS stressed conditions"; L303, "DOA10"; L427, "gDNA"; L429, "ACT1". Please indicate full name when necessary.

Response: Thanks for the reviewer's comments. We have proof-read the manuscript thoroughly to define all abbreviations for the first use. SDS was firstly defined in Line 90, DOA10 in Line 304-305, *ACT1* in Line 438, and some other abbreviations in the revised manuscript.

Reviewer 2

Comment 1: Line 176: “The sterol homeostasis between LDs and cytoplasm....” - Esterified sterols are found in LDs and non-esterified sterol in cellular membranes – but not in the cytoplasm!

Response: Thanks for the reviewer's comments. We have corrected it as “The sterol homeostasis between LDs and cellular membranes” in Line 68 and Line 178 of the revised manuscript.

Comment 2: Sometimes the wording is wrong, e.g.:

Line 220: “...inducible promoters might resulted in..” – might result in

Line 266: “... which could be severed as...” – which could serve as

Some gene names are not written in italic – please correct this.

Response: Thanks for the reviewer's comments. We have corrected all grammar errors pointed by the reviewer as well as some other minor errors in the revised manuscript.

Comment 3: Figure 2B, right panel: As indicated, this panel shows the titer of 24-epi-ergosterol in different strains. However, according to the text in the manuscript (line 146 and 147), the strain YQP1 harboring the plasmid pRS42H is producing high levels of ergosterol and no 24-epi-ergosterol! This must be corrected in the Figure 2B.

Response: Thanks for the reviewer's comments. Figure 2B has been revised to show ergosterol and 24-epi-ergosterol correctly (Figure 2B of the revised manuscript).

Comment 4: Table S3: Please indicate what the colors in the table stand for

Response: Thanks for the reviewer's comments. Originally, we used the color density

to indicate the relative transcriptional level of promoters under different conditions. As the promoter strengths have been detailed using numbers, we think it unnecessary to keep the color any more. Therefore, we removed colors in Supplementary Table 4 of the revised SI. So it is with Supplementary Table 1 of the revised SI.

Reviewer 3

Comment 1: My main concern is that the enzyme directed evolution may have reduced the stereoselectivity of the enzyme because there is no pressure in the selection to prevent the enzyme from losing stereoselectivity since production of either 24-epi-ergosterol and ergosterol should be sufficient for survival. The authors show HPLC and LC-MS data in the SI for the WT enzyme that nicely demonstrate that the WT enzyme is stereoselective and produces 24-epi-ergosterol and not ergosterol. But the same data are not shown for the final engineered enzyme Ar207. Therefore, it would be helpful to see the equivalent of Figure S3 and Figure S4 for the yeast strain containing Ar207 as well as the final best engineered strain YQE734 compared to an ergosterol standard to ensure that these strains are selectively producing 24-epi-ergosterol over ergosterol.

Response: We appreciate the reviewer's valuable comments. HPLC profile in Supplementary Fig. 4B is for the yeast strain containing engineered enzyme Ar207 and the stereoselectivity was found to be not affected. In addition, we have provided HPLC profile for the final best engineered strain YQE734 and ergosterol standard in Supplementary Fig. 15 of the revised SI, which clearly showed that to ensure that the strain is selectively producing 24-epi-ergosterol other than ergosterol.

Comment 2: Another minor issue is that while all experiments were done in triplicate, there aren't any statistics reported for any of the results. In some cases the significance of differences reported are clear but other cases they are not. For example, some of the mutant enzymes (especially the single mutants E61K and Q519R) do not appear to be significantly better than WT but they are all reported as improvements in the text. I recommend adding indications of significance on the plots comparing titers or OD₆₀₀

so it is easier to tell what differences are statistically significant, and mentioning p-values, confidence intervals, or some other measure of significance in the text when claiming significant improvements.

Response: Thanks for the reviewer's suggestion. We have added indications of statistical significance in all figures of the revised manuscript and SI (Figure 2, Figure 3, Figure 4, Supplementary Fig. 6, Supplementary Fig. 7, Supplementary Fig. 11, and Supplementary Fig. 13), in the form of "*" or p-values. As for the single mutants E61K and Q519R, although they do not appear to be significantly, we think it may improve the activity of DWF1 when combined with other mutations. For example, Ar203 performed much better than Ar101, indicating that E61K was beneficial for DWF1 activity when combined with Y143G and S306P.

- 1 Sun, Z. J. *et al.* Combined Biosynthetic Pathway Engineering and Storage Pool Expansion for High-Level Production of Ergosterol in Industrial *Saccharomyces cerevisiae*. *Front Bioeng Biotech* **9** (2021).

Reviewers' Comments:

Reviewer #1:

Remarks to the Author:

The authors have done a good job of addressing the various concerns raised by the reviewers.

Reviewer #2:

Remarks to the Author:

All my concerns have been addressed in the revised manuscript.

The manuscript is now ready for publication.

Reviewer #3:

Remarks to the Author:

The authors have addressed all the major concerns from my first review. I suggest that Supplementary Figure 4 be referenced in the results section that discusses Ar207. I missed that Supplementary Figure 4 contained data related to Ar207 in my initial review because it was not referenced in association with Ar207.

I have no other concerns or suggestions.

Reviewer #1 (Remarks to the Author):

The authors have done a good job of addressing the various concerns raised by the reviewers.

Thanks for the reviewer's positive comments and all the suggestions to improve our manuscript.

Reviewer #2 (Remarks to the Author):

All my concerns have been addressed in the revised manuscript. The manuscript is now ready for publication.

Thanks for the reviewer's positive comments and all the suggestions to improve our manuscript

Reviewer #3 (Remarks to the Author):

Comment 1: The authors have addressed all the major concerns from my first review. I suggest that Supplementary Figure 4 be referenced in the results section that discusses Ar207. I missed that Supplementary Figure 4 contained data related to Ar207 in my initial review because it was not referenced in association with Ar207. I have no other concerns or suggestions.

Thanks for reviewer's valuable suggestion. We have referenced Supplementary Fig. 4b in association with Ar207 in Line 169-170 of the revised manuscript.